# Mapping Risk of Malaria as a Function of Anthropic and Environmental Conditions in Sussundenga Village, Mozambique

**DOI:** 10.3390/ijerph18052568

**Published:** 2021-03-05

**Authors:** João L. Ferrão, Dominique Earland, Anísio Novela, Roberto Mendes, Marcos F. Ballat, Alberto Tungaza, Kelly M. Searle

**Affiliations:** 1Instituto Superior de Ciências e Educação a Distância, Beira 2102, Mozambique; 2School of Public Health, University of Minnesota, Minneapolis, MN 55455, USA; earla001@umn.edu (D.E.); ksearle@umn.edu (K.M.S.); 3Direcção Distrital de Saúde de Sussundenga, Sussundenga 2207, Mozambique; novela.anisio@hotmail.com; 4Centro de Informação Geográfica-Faculdade de Economia da UCM, Beira 2102, Mozambique; rmendes@ucm.ac.mz; 5Faculdade de Ciência de Saúde da UCM, Beira 2102, Mozambique; mballat@ucm.ac.mz; 6Faculdade de Engenharia da UCM, Chimoio 2203, Mozambique; atungadza@gmail.com

**Keywords:** mapping, risk, malaria, Sussundenga village

## Abstract

Mozambique is a country in Southern Africa with around 30 million inhabitants. Malaria is the leading cause of mortality in the country. According to the WHO, Mozambique has the third highest number of malaria cases in the world, representing approximately 5% of the world total cases. Sussundenga District has the highest incidence in the Manica province and environmental conditions are the major contributor to malaria transmission. There is a lack of malaria risk maps to inform transmission dynamics in Sussundenga village. This study develops a malaria risk map for Sussundenga Village in Mozambique and identifies high risk areas to inform on appropriate malaria control and eradication efforts. One hundred houses were randomly sampled and tested for malaria in Sussundenga Rural Municipality. To construct the map, a spatial conceptual model was used to estimate risk areas using ten environmental and anthropic factors. Data from Worldclim, 30 × 30 Landsat images were used, and layers were produced in a raster data set. Layers between class values were compared by assigning numerical values to the classes within each layer of the map with equal rank. Data set input was classified, using diverse weights depending on their appropriateness. The reclassified data outputs were combined after reclassification. The map indicated a high risk for malaria in the northeast and southeast, that is, the neighborhoods of Nhamazara, Nhamarenza, and Unidade. The central eastern areas, that is, 25 de Junho, 1 and 2, 7 de Abril, and Chicueu presented a moderate risk. In Sussundenga village there was 92% moderate and 8% high risk. High malaria risk areas are most often located in densely populated areas and areas close to water bodies. The relevant findings of this study can inform on effective malaria interventions.

## 1. Background

Mozambique is a country in Southern Africa with around 30 million inhabitants. Malaria is the leading cause of mortality in the country. According to the WHO, Mozambique has the third highest cases worldwide, approximately 5% of the world cases [1,2]. In Mozambique malaria is endemic across the country, ranging from hyper-endemic in areas along the coast, meso-endemic in inland flatlands, and hypo-endemic areas in the highlands. Malaria transmission occurs all year round, peaking during the rainy season. *Anopheles funestus* is the most frequent, with 90% of the cases, followed by *Plasmodim ovale* with 9%, and *Plamodium malariae* with 1% of the cases [3].

Manica Province, located in the central region of Mozambique, has the fourth largest number of cases among all provinces in Mozambique. In 2019 the province recorded 821,775 malaria cases. With Manica Province, Sussundenga District had an increased number of malaria cases and in the first six months of 2020, recorded 63,526 cases accounting for around 19% of the province cases (out of 12 districts) [4].

Several factors contribute to malaria endemicity, including climate and environmental conditions such as increased temperatures and rainfall patterns, favorable places for vector development, and sociodemographic and economic conditions. Prior study in Chimoio, Mozambique [5] indicated that environmental conditions, account for 73% of malaria cases while, Global Fund [6] indicated a figure of 90%. A study in Sussundenga reported that social determinants contribute to 15% of the positive malaria cases.

The spatial distribution of transmission intensity and associated variables have become an urgent requirement, especially in endemic areas. GIS is an innovative technique to understand malaria spatiality and identify high-risk zones to develop appropriate interventions [7]. A malaria risk map does not exist for the Sussundenga village, as well as local scale maps to localize risk areas based on the distance between the vector breeding sites and host households, the vector dispersion, and case clusters [8]. The maps could also be useful long term for supporting elimination efforts. The objective of this study was to develop a malaria map of for Sussundenga Village in Mozambique, identifying zones at risk to inform malaria eradication efforts, and collect environmental, anthropic, and clinic data.

## 2. Methods

### 2.1. Study Area

Sussundenga village (Figure 1) is located in Sussundenga District, 42 km from Chimoio, the capital of Manica Province, around 45 km from the Zimbabwe border.

The rural and agrarian region is about 156.9 square kilometers and has 41,354 inhabitants. The village is administratively divided into 17 residential areas called “Bairros” (Figure 2) [9].

There are two seasons, the rainy season starts in November and ends in March, the dry season is from April to October. The average rainfall is 1067.6 mm, which varies significantly in quantity and distribution within and between years. The average highest temperatures are in October and November (30 °C), the lowest in July (12 °C) [10].

The predominant vegetation consists of deciduous forests, evergreen forests, prairie, shrubs, and savanna. The hydrographic network is comprised of 6 main rivers, Revué, Munhinga, Mussapa, Lucite, Chicueu, and Muzória with permanent flow, and several waterbodies built for irrigation. The village has one public health center and the majority of residents have low access to health services. Bairros 25 Junho 1, 2, and Chicueu have a health center in close proximity [9].

### 2.2. Data Collection and Analysis

Spatial, anthropic, and environmental variables were used to model the risk map. A four-step strategy was applied to develop the map as described by Ferrao et al. [11].

The factors of risk used can be seen in Figure 3 as well as the included population density (inhabitants/km^2^), point prevalence per Bairro (%), average temperature (°C), precipitation (mm), altitude (meters), slope (degrees), NDVI (Normalized digital index), distance of the road (km), distance to waterbodies (km), and Land usage and cover (LULC).

Ten risk factors were used and Table 1 demonstrates the risk factors of malaria, with the weights and classes of anthropic and environmental conditions.

#### 2.2.1. Point Prevalence

One Hundred houses were randomly sampled and tested for malaria in Sussundenga Village using a malaria rapid diagnostic test (RDT), RightSign Biotest^®^ (Biotest, Hangzhou Biotech Co, China). The field team visited communities (Bairros) in different locations in the village, based on samples of the individual at malaria risk in each Bairro. They recorded the individual test result and, each individual positive case was used to calculate the point-prevalence, that is, the proportion of individuals in a population that has the disease at a single point in time, regardless of the duration of time that the individual might have that disease. This was calculated by dividing the number of positive cases per residential area by the total population of the residential area and multiplying by 100 [12] using the following formula:(1)$$Point\;Prevalence\;\left(\%\right)=\frac{Persons\;malaria\;positive}{Bairro\;Population\;during\;the\;period}-100$$

#### 2.2.2. Average Temperature (°C)

Long term minimum, maximum temperature as presented in Figure 4 was extracted from WorldClim-Global Climate Data (European Environmental Agency), from 1970 to 2000 [13]. The average temperature was calculated and thematic map was produced using ArcGis 10.7.1 (Esri, Redlands, CA, USA). [14].

#### 2.2.3. Precipitation (mm)

Precipitation data as presented in Figure 4 were extracted from the Bioclim (WorldClim) from 1970 to 2000 [13], processed at Diva GIS 7.5 (Centro International da Papa & Food Agricultural Program, Lima-Peru, Rome-Italy) [15] and the thematic map produced.

#### 2.2.4. Altitude (Meters)

A digital elevation model (DEM) [16] with a resolution of 30 × 30 m was used to estimate the altitude and thematic map.

#### 2.2.5. Slope (Degrees)

The slope was derived from the 30 m×30 m digital elevation model [16], which was obtained from the ArcGIS and thematic map spatial analysis tool.

#### 2.2.6. Land Cover and Land Use (LULC)

Land cover and land use data were retrieved from the most recent Landsat 8 satellite image (September 2019), Appendix A [17] and thematic map. The image was reclassified into different LULC classes using the manual training sampling technique and the maximum likelihood algorithm. The LULC classes were as follows:Agricultural crop area, grass. and water body.Shrubland and mosaic cover vegetationForest, bare and urban settlement areas [18].

#### 2.2.7. Distance to the Road (km)

The Euclidean distance to the nearest road was calculated using ArcGIS, classifying a 2019 Landsat image, 30 m × 30 m and thematic map. Distances from road locations were calculated using the distance measurement function in the ArcGIS 10.7.2 software.

#### 2.2.8. Distance to Waterbodies (km)

The distance to the nearest body of water was calculated with ArcGIS 10.7.2, classifying a 2019 Landsat image of 30 m 30 m for the water and undefined zones [19] and the thematic map.

#### 2.2.9. Vegetation Index by Normalized Differences (NDVI)

Vegetation vigor is indicated by NDVI, and greater amounts of green vegetation in the soil indicate higher NDVI. Non-vegetation classes are generally lower than the vegetation classes in NDVI and the thematic map. The resulting expression of NDVI is as follows:(2)$$\mathrm{NDVI}=\left(\frac{\mathrm{NIR}-\mathrm{RED}}{\mathrm{NIR}+\mathrm{RED}}\right)$$
where:

NIR is the reflectance in the near infrared band.

RED is the reflectance in the red band [20].

NDVI was taken from the Landsat image.

#### 2.2.10. Determining Risk Factor Weights (Analytical Hierarchical Process)

Analytical Hierarchical Process (AHP), a simple way to assign relative weights for different factors through applying pairwise comparison was applied [21,22]. In pairwise comparisons each factor is evaluated according to a scale ranging from 1 to 9 (Table 2). As a result of the pairwise comparisons process, a reciprocal matrix was produced, where each factor in the matrix represented the dominance of a certain factor over another in terms of their contribution to malaria risk. Thereafter, each factor in the reciprocal matrix was divided by the sum of its column. Finally, the weight of each factor was calculated as an average across the rows [23]. The true consistency ratio was used after the pairwise matrix derivation, which was calculated by dividing the consistency index for the set judgments by the index for the corresponding random matrix. Saaty suggests that, if that ratio exceeds 0.1, the set of judgments may be too inconsistent to be reliable [24].

#### 2.2.11. Mapping Risk of Malaria

Figure 5 represents the schematic representation of the data flow and analysis to generate a malaria risk map in the Sussundenga Village.

In this step weighted overlay analysis, a modeling method for suitability, that analyses multiclass maps based on the relative importance of each thematic layer and layers class was used [25]. Raster layers were overlayed and multiplied by their assigned weight. Raster layers were overlayed and multiplied by each raster cell’s suitability value and its layer weight to derive a suitability value (Table 2), as presented in the following formula:(3)$$S=\;\frac{\sum \mathrm{WiSij}}{\sum \mathrm{Wi}}$$
where:

*Wi* is the weight of the *i*-th factor map,

*Sij* is the *i*-th spatial class weight of *j*-th,

*S* is the spatial unit value in the output map [26]. As a result, a new raster surface was generated representing different levels of malaria risk based on the anthropic and environmental conditions.

#### 2.2.12. Accuracy Assessment of the Produced Map

The intention in this step was to evaluate the accuracy of the produced risk map. Data analysis of the 2019 record books was also carried out to access location of the malaria patients as a tool to check for accuracy. The risk map was also discussed with health practitioners and malaria program officers in Sussundenga to assess accuracy.

## 3. Results

The average temperature ranges from 21.07 °C to 21.74 °C. Precipitation ranges from 1092.01 mm to 1124 mm. The altitude ranges from 534 m to 711 m. The slope varies from 0% to 24.32% (Figure 6).

Figure 7 shows the maps of water bodies (DTWB), distance to the road (DTR), population density (PD), malaria point prevalence (MPP), and normalized difference index (NDVI). The DTWB varied from 0 to 4116.06 m, the DTR varied from 0 to 8153.21 m, the PD varied from 1639 to 41,770 inhabitants/km^2^. MPP varied from 0 to 47.29% and NDVI from −0.02 to 0.49.

Table 3 represents the comparison matrix of malaria risk factors used for weighting. A value of 1 means that the comparison factors have the same weight and that they equally affect malaria occurrence. A value of five means that the factor in the column has five times the risk of malaria occurrence than the comparison in the row.

The weights of each factor used for the spatial model to produce the malaria risk map were: average temperature (22.6%). precipitation (20.0%). altitude (12.7%). distance to the water body (9.36%). slope (8.28%). land use and cover (7.69%). population (6.46%). distance from the road (4.86%). malaria prevalence (4.73%). NDVI (3.30%). The consistency ratio for the pairwise matrix was 0.08.

Figure 8 shows the maps of average temperature. precipitation. altitude. slope and LULC. In terms of temperature. 100% of the Sussundenga village area was at low risk. For precipitation. the entire village (100%) had a high risk. For altitude. 100% of the area of the village of Sussundenga was at high risk. For slope. 0.21% had low risk. 14.51% had moderate risk. and 85.28% high risk. For LULC. 30.74% of the area had a low risk. 7.73% a moderate risk. and 61.53% a high risk.

Figure 9 shows the maps of DTWB. DTR. population density. prevalence of malaria. and use and normalized difference vegetation index (NDVI). For DTWB. 28.52% of the area had low risk. 48.68% moderate risk. and 22.80% high risk. For DTR. 58.19% had low risk. 21.96% had moderate risk. and 19.85% high risk. For population density. 70.57% had a low risk. 24.13% had a moderate risk. and 5.30% a high risk. For malaria prevalence. 92% had low risk. 3% had moderate risk. and 5% high risk. For NDVI. 0.5% had low risk. 90% had moderate risk. and 9.5% high risk.

Figure 10 represents the malaria risk map. Neighborhoods of Nhamaraza. Nhamarenza. Chicueu. and Unidade have higher incidence of malaria in the northeast and southeast areas. In contrast. the central eastern part. had a lower risk of malaria. in the neighborhoods 25 de Junho 1. 25 de Junho 2. and 7 de Abril.

Table 4 represents the area under different malaria risk in Sussundenga village. Around 8% of the area is high malaria risk and 92% is moderate malaria risk in Sussundenga village.

### Accuracy Check

Comparing the map results and the data extracted from the book records (Appendix B) it can be noted that only one neighborhood (14%) failed the prediction. The consistency ratio for the pairwise matrix was 0.08.

## 4. Discussion

There are currently no studies available on the mapping risk of malaria in Sussundenga village. In this study precipitation. slope. and LULC were related to malaria risk. Similar results have been reported in Chimoio [11]. in Zimbabwe [22]. Burundi [23]. Zambia [27]. Ghana [28]. Colombia [28,29]. and China [30]. Rainfall increases the presence of stagnant water for malaria vectors for breeding and a weekly precipitation of over 10 mm will propitiate mosquito development [31]. Sussundenga has an average weekly precipitation of 22 m. from June to September the weekly precipitation is below 10 mm [10]. which may explain the high malaria risk in Sussundenga. Flat landscape areas are more prone to accumulate water from rain which increases the malaria risk [32] and may explain the increased risk in Sussundenga.

In this study. LULC seems also to play an important role in malaria occurrence. Modified man-made landscape is conducive to malaria vectors. which could be attributed to the elevated population. [8,28].

The malaria risk map in this study. suggests that places near bodies of water and higher population density have increased malaria risk compared to the other areas. Large and small dams can increase malaria cases in villages that are located near them. since dams create additional breeding habitats for vectors and higher malaria transmission [33,34]. In Maputo. Mozambique. the risk of malaria was 6.2 times greater for individuals living less than 200 m from the breeding sites than that of individuals living 500 m or more from the breeding sites [34]. Rapid urbanization of areas within villages or on the outskirts of existing urban centers is common and often developed in without proper planning. Conditions are crowded and developers tend to dig pits to extract stones and soil for house construction. creating numerous breeding sites for mosquitoes. [8,28]. In densely populated areas. mosquitoes do not have to travel far to feed [35].

In this study. Sussundenga village had malaria risk ranging from high to moderate. Similar results were reported in Chimoio. Mozambique [11]. The consistency ratio for the pairwise matrix was 0.08. which is well below Saaty’s suggested 0.1 ratio to delineate a reliable result [24]. The map accuracy in this study was 86% and similar results were reported in Swaziland [36]. In Madagascar [37] an accuracy of 74% was reported [38].

## 5. Limitations of the Study

This paper was an attempt to validate a previous study carried out in Chimoio in 2018 and which would have been expected to generate a predictive model that could predict malaria prevalence without the necessity of prevalence as an input.

## 6. Conclusions

Sussundenga village had 8% with high risk and 92% moderate risk. High malaria risk areas are most often located in densely populated areas and areas close to water bodies. The relevant findings of this study can inform on effective malaria interventions.

## Figures and Tables

**Figure 1 ijerph-18-02568-f001:**
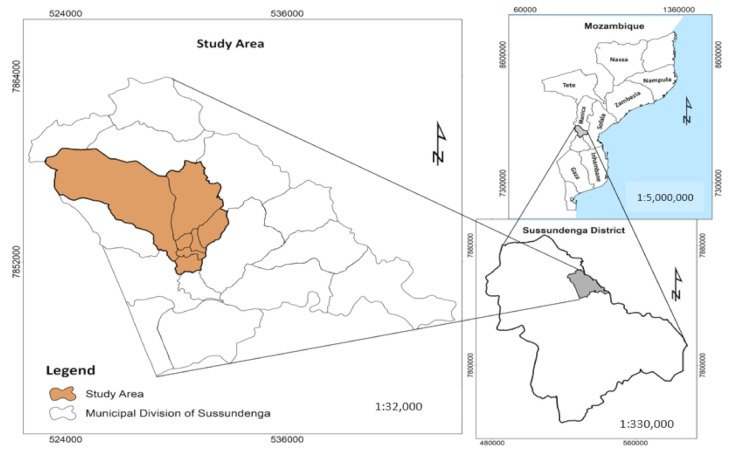
Map of Mozambique, Sussundenga Village and the study area.

**Figure 2 ijerph-18-02568-f002:**
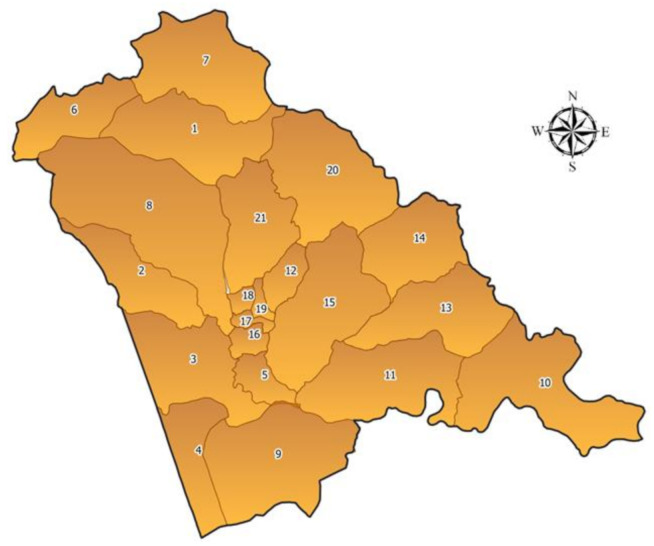
Sussundenga village map and Bairro’s. 1. Nhamatiquite, 2. Samora Machel, 3. Bapua, 4. Congrosso, 5. Chizizira, 6. 3 de Fevereiro, 7. Mussacumbira, 8. Nhamarenza, 9. Dhoa, 10. Chissamba, 11. Nhaguzue, 12. Chissanba, 13. Chipendeque, 14. Tave, 15. Muzoria, 16. Unidade, 17. 7 de Abril, 18. 25 de Junho 2, 19. 25 de Junho 1, 20. Nhamawaia, e 21. Nhamezara.

**Figure 3 ijerph-18-02568-f003:**
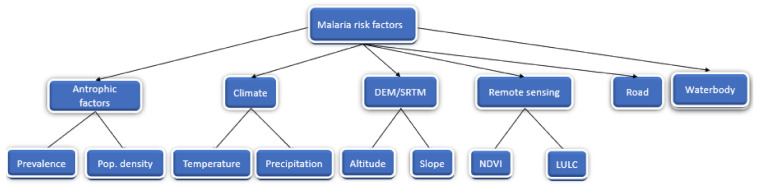
Representation of malaria risk factors and their flow for malaria risk map generation for Sussundenga Village. DEM/SRTM = Digital elevation model/Shuttle Radar Topographic Mission, NDVI = Normalized difference vegetation index, LULC = Land use and land cover.

**Figure 4 ijerph-18-02568-f004:**
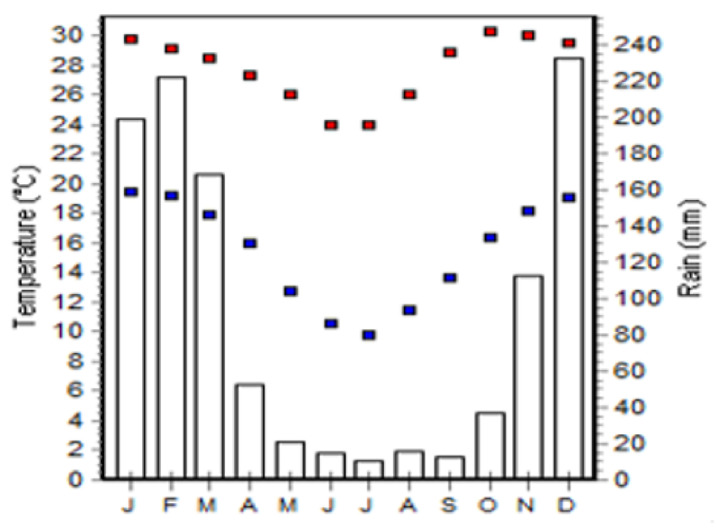
Long term minimum and maximum temperature and, precipitation for Sussundenga 1970/2000. Source: Worldclim (European Environmental Agency).

**Figure 5 ijerph-18-02568-f005:**
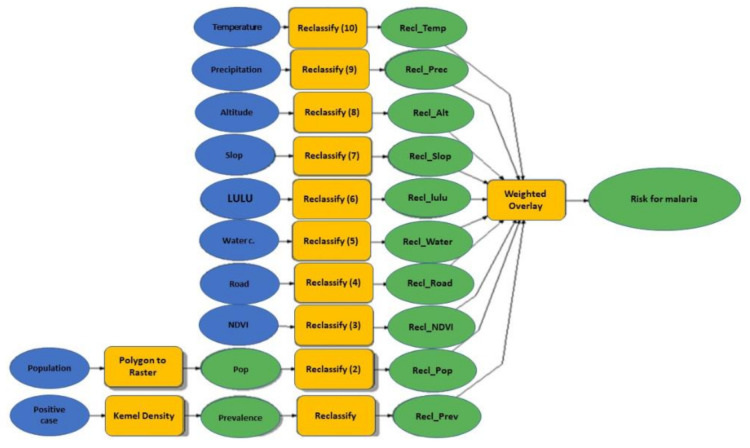
Schematic representation of the data flow and analysis to generate a malaria risk map in the Sussundenga Village.

**Figure 6 ijerph-18-02568-f006:**
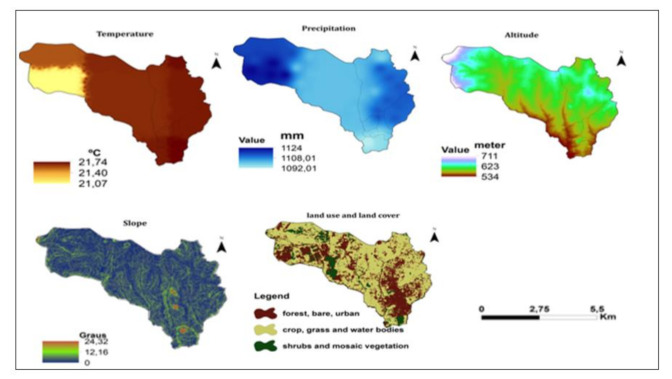
Presents the map of average temperature, precipitation, altitude, slope, and land in Sussundenga Village.

**Figure 7 ijerph-18-02568-f007:**
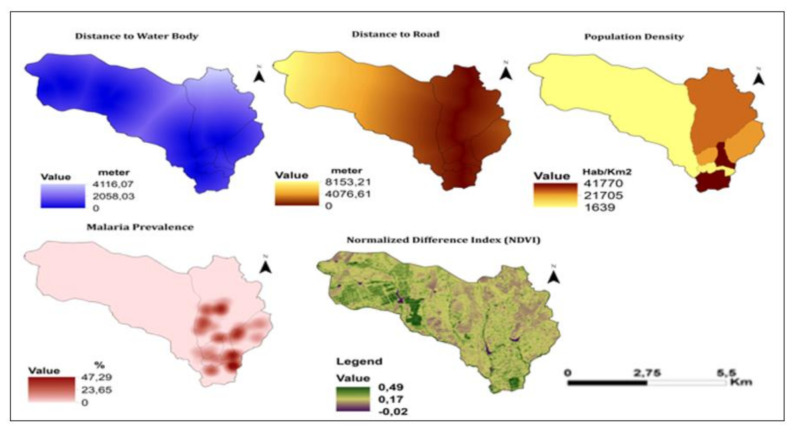
Maps of water bodies (DTWB). distance to the road (DTR). population density (PD). malaria point prevalence (MPP) and normalized difference index (NDVI) in Sussundenga Village.

**Figure 8 ijerph-18-02568-f008:**
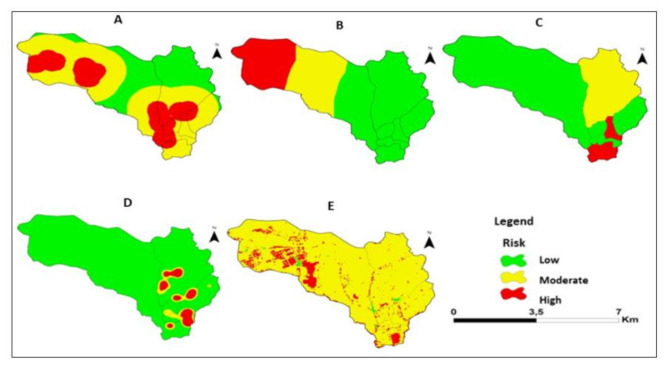
Risk for malaria. (**A**). Temperature. B) Precipitation. (**C**). Altitude. (**D**). Slope. (**E**). LULC for Sussundenga Village.

**Figure 9 ijerph-18-02568-f009:**
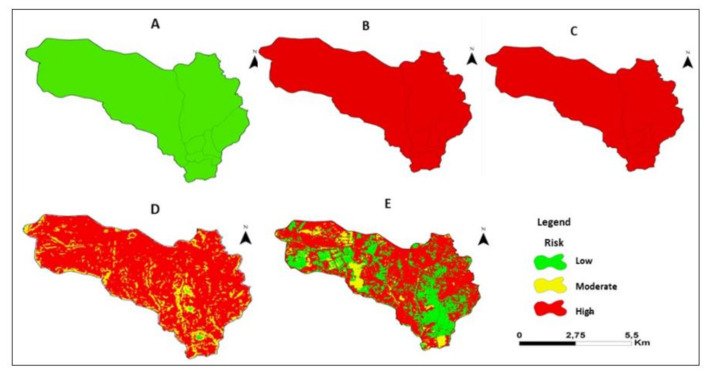
Risk for malaria. (**A**). Distance to waterbodies. (**B**). Distance to roads. (**C**). Pop. Density. (**D**). Malaria prevalence. (**E**). NDVI for Sussundenga Village.

**Figure 10 ijerph-18-02568-f010:**
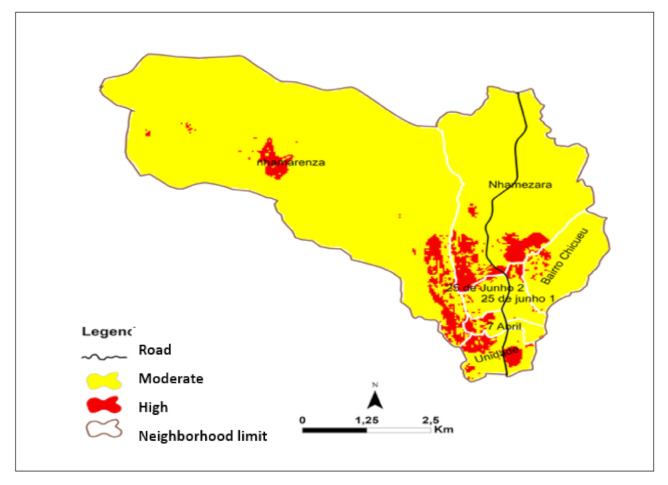
Malaria risk map for Sussundenga Village.

**Table 1 ijerph-18-02568-t001:** Malaria risk factor, weights, and classes of anthropic and environmental conditions.

Risk Factor	Weight %	Class	Rank	Risk Degree
Average Temperature °C	22.4	22–32	3	High
		>32	2	Moderate
		<22	1	Low
Precipitation (mm)	20.8	>700	3	High
		450–700	2	Moderate
		<450	1	Low
Altitude (mm)	10.4	<200	3	High
		201–500	2	Moderate
		>500	1	Low
Slope (degrees)	7.3	0–5	3	High
		5–15	2	Moderate
		>15	1	Low
LULC	8.2	Agric. crop area, grass and water body.	3	High
		Shrubland & mosaic cover vegetation	2	Moderate
		Forest, bare and urban settlement	1	Low
DTWB (km)		<500	3	High
		500–1500	2	Moderate
		>1500	1	Low
DTR (km)	3.8	>5	3	High
		2.5–5	2	Moderate
		<2.5	1	Low
Pop. Density	5.1	>9000	3	High
		6001–9000	2	Moderate
		<6000	1	Low
Prevalence (%)	5.1	>21	3	High
		14–21	2	Moderate
		<14	1	Low
NDVI	4.7	0.255–0.986	3	High
		0–0.25	2	Moderate
		−0.288–0	1	Low

LULC = Land use and land cover, DTWB = Distance do Waterbody, DTR = Distance do Road, Pop. = Population, NDVI = Normalized difference vegetation index. Source: dated from: Ferrao et al. Mapping and Modelling Malaria Risk Areas Using Climate, Socio-Demographic and Clinical Variables in Chimoio, Mozambique.

**Table 2 ijerph-18-02568-t002:** Scale to develop the pairwise comparison matrix.

1	Equal importance	Two factors also contribute equally to the objective.
3	Moderate importance	Experience and judgment slightly favor one factor in relation to the other.
5	Much more important	Experience and judgment strongly favor one factor over the other
7	Very important	Experience and judgment very strongly favor one over the other factor.
9	Absolutely important	The evidence favoring one over the other is the highest possible validity
2,4,6,8	Intermediate values	When compromise is needed

**Table 3 ijerph-18-02568-t003:** 10 × 10 Comparison matrix of Risk Factors used in the study.

Risk Factor	T Average	Prep	Alt	SLP	LULC	DTWB	DTR	Pop	Prev	NDVI
T average	1.00	1.00	3.00	4.00	4.00	2.00	7.00	4.00	4.00	5.00
PP	1.00	1.00	3.00	4.00	3.00	1.00	7.00	4.00	4.00	3.00
Alt	0.33	0.33	1.00	3.00	3.00	1.00	4.00	2.00	2.00	3.00
SLP	0.25	0.25	0.33	1.00	1.00	2.00	1.00	3.00	2.00	2.00
LULC	0.25	0.33	0.33	1.00	1.00	2.00	2.00	3.00	1.00	1.00
DTWB	0.50	1.00	1.00	0.50	0.50	1.00	2.00	3.00	3.00	2.00
DTR	0.14	0.14	0.25	1.00	0.50	0.50	1.00	1.00	1.00	2.00
Pop den	0.25	0.25	0.50	0.33	0.33	0.33	1.00	1.00	2.00	4.00
Prevalence	0.25	0.25	0.50	0.50	1.00	0.33	1.00	0.50	1.00	2.00
NDVI	0.20	0.33	0.33	0.50	1.00	0.50	0.50	0.25	0.50	1.00

T average = mean temperature. Prep = precipitation. Alt = Altitude. SLP = Slope. LULC = Land use and land cover. DTWB = Distance to waterbodies. DTR = Distance to Road. Prev = Prevalence.

**Table 4 ijerph-18-02568-t004:** Areas under different malaria risk in Sussundenga Village.

Value	Risk Classification	Area (Hectare)	Percentage
1	Hight	244.44	7.59
2	Moderate	2972.34	92.41
	Total	3216.78	100

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
