# Peer review of "Mapping Risk of Malaria as a Function of Anthropic and Environmental Conditions in Sussundenga Village, Mozambique"

_ijerph, 2021, doi:10.3390/ijerph18052568_

Round 1
Reviewer 1 Report
Ferrao et al have presented an interesting and nicely analyzed risk assessment of malaria in Sussendenga Village, Mozambique. This paper replicates a similar study performed for Chimoio that the group reported on a few years prior. While overall the paper is in general well written there are multiple areas of improvement.
Methods:
Definitions of point-prevalence: the authors do not describe whether each household was considered as a single entity or whether each individual tested within the household was considered separately. Considering the flight range of Aedes mosquitoes is short, it is important to carefully describe which option was chosen and why.
The authors state that risk factor weights were assessed using a 5 degree scale, however, Table 2 presents a 9-point scale. The correct weighting would appear to be based on a 9-point scale since intermediate values (2,4,6,8) are allowed.
Results
The authors show that only one neighborhood failed to correctly predict malaria prevalence. What are the standard thresholds to accept this risk map?
Relevance of the study
The methodology presented both in this paper and in the 2018 paper rely on an iterative approach to create a model of predictive disease prevalence that relies on the measured prevalence as a model input. This would appear to result in a circular analysis. The 2018 paper presents a nice analytic analysis, and would have been expected to generate a predictive model that could predict malaria prevalence without the necessity of prevalence as an input. The data from Sussendenga could thus be used as a measure of the validity of the prior model to determine the model's limitations and where improvements should be made. The authors should discuss this as a limitation and how to create a purely predictive model.
Other
There are numerous grammatical and spelling errors
The language is not entirely English
Tables and Figure numbers need to be carefully edited for accuracy. For example, three of the figures are labelled as Figure 8.
Abbreviations used in the paper should be consistent throughout.
Author Response
Reviewer 1

Reviewer 2 Report
Journal: Int. J. Environ. Res. Public Health
Manuscript ID: ijerph-1084725
Type of manuscript: Article
Title: Mapping risk of malaria as a function of anthropic and environmental conditions in Sussundenga village, Mozambique
Authors: Joao Ferrao, Dominique Earland, Anisio Novela, Roberto Mendes, Antonio Tungadza, Marcos Ballat, Kelly Searle
General Comments
Malaria is known to be a disease characterized by patchy distribution even in endemic areas. Studies that describe malaria risk also at local level can provide relevant information for appropriate control plans and can contribute to reduce effectively the burden of this disease in endemic areas.The present manuscript submitted by Joao Ferao et al, provides a map of malaria risk for Sussundenga Village in Mozambico, considering different environmental and anthropic factors present in the area.
Overall, the paper deserves to be published in Journal: Int. J. Environ. Res. Public Health, however I would like to point out to the authors that some minor revisions are needed, as described in the specific comments reported below. There are several typos in the text and the references should be carefully checked. A language revision is recommended before being published.
Specific comments
Background
Page 1:
correct the species name as below:
“The main vectors are Anopheles funestus and Anopheles gambiae, and Plasmodium falciparum is the most frequent, 90 % of cases, followed by Plasmodim ovale 9 % and Plamodim malariae 1 % [3].”
Page 2:
it would be better to write:
…”favorable places for vector development and sociodemographic and economic conditions.”
correct as below:
“…..Global Fund [6] indicated a figure of 90 %.”…
Methods
Page 3:
please, re-write better the caption of figure 1
correct as below
“…the rain starts in November and ends in March, …”
Page 4:
in the text and table 1 write correctly: “…°C…”
In the flow chart the arrow connecting the Malaria risk factors box and the DEM-SRTM box is missing
Page 5:
correct as below
“… maximum temperature as presented in figure 4 was…”
Page 6:
correct as below
“Precipitation data as presented in figure 4,…”
add legend in Figure 4
remove the comma caption
Results
Page 9:
remove double space
…”distance to the road (DTR), population density…”
please, re-write better the caption of figure 6, also referring to the study area (Sussundenga Village) and insert in the text the reference to figure 6
Page 10:
in Figure 7 caption, remove the spacing
“… distance to the road (DTR),…”; add the study area (Sussundenga Village)
in Table 3, correct “Prevalência” vs “Prevalence”
in the text, add
“…the population density (6.46%),…”
Page 11:
in Figure 8 caption, add the study area (Sussundenga Village)
in the legend, correct “risco” vs “risk” and “higt” vs “high”
Page 12:
correct the number of the figure, “8” vs “9”
add the letters to each map
correct “Pop. Density” vs “Pop. density”
in the caption, add the study area (Sussundenga Village)
in the legend correct “risco” vs “risk” and “higt” vs “high”
Page 13:
correct the number of the figure, “8” vs “10”;
in the legend correct “higt” vs “high”
in the caption of Table 4 correct “Malaria Risk” vs “malaria risk”; in the table correct “hight” vs “high”
In the text, add the parenthesis: “….one neighborhood (14 %) failed the prediction.”
Page 14
Appendix 1
Please, use English language in the caption
References
Many references are incomplete, such as number 11:
Ferrao JL, Niquisse SA, Mendes JM, Painho M. Mapping and Modelling Malaria Risk Areas Using Climate, Socio-Demographic and Clinical Variables in Chimoio, Mozambique. International Journal of Environmental Research and Public Health. 2018.
should be as reported below:
Int J Environ Res Public Health. 2018 Apr; 15(4): 795. Published online 2018 Apr 19. doi: 10.3390/ijerph15040795
Review carefully also the reference numbers: 12, 21,22,24,25,29,30,32,33,34,36,37,39.
Author Response
Reviwer 2

Reviewer 3 Report
The authors map out risk factors for malaria in Sussundenga District found in the province of Manica in Mozambique. This study could be beneficial, but authors do not clearly state how this work will be beneficial. The methods are well detailed but there are major points of concern is the way the data is presented.
- There is lack of details in the way the methods and discussion sections are presented.
- It would be beneficial for the authors to indicate how mapping risk factors for malaria in this areas will be beneficial in the long term.
- In some places the authors indicate that Sussundenga is a district while in some places, they indicate that it is a village. It's better to differentiate between the two or state the reason for the change in these descriptions as a district is different from a village.
- There is no consistency in figure formatting. Some are blurry, some have bold fonts while others not, some do not have enough details to help the reader understand
- The formatting of tables could also be improved
- Generally the discussion section is poorly organized which makes it hard for the reader to know the take home message or the importance of the study and how it contributes to the field
- There are a lot of grammatical errors and poor sentence structure that could benefit from good editing.
Author Response
Reviewer 3

Round 2
Reviewer 3 Report
The authors have addressed most of this reviewers concerns but there are still several dramatically errors. Even though the authors indicate that a native English speaker will revise the manuscript, there is no evidence that this happened as some of the errors that were in the initial submission are still in this revised version. Some of the added information has grammatical errors and abstract seem to have different font than the paper itself. Below is a list of few sentence with wrong sentence structure and/or grammatical errors.
- Areas with increased malaria risk were located in highly populated and closest to bodies of water.
- Anopheles funestus and Anopheles gambiae, and Plasmodium falciparum is the most frequent, 90 % of cases, followed by Plasmodim ovale 9% and Plamodium malariae1 % [3].
- Prior study in Chimoio, Mozambique [5] indicated that environmental conditions, account for about73 % of malaria cases while, Global Fund [6] indicated a figure of 90 %.[6] indicated a figure of 90 %.
Author Response
Reviewer 3.
Thanks so much for the comments. Minor revisions were carried out in the abstract and the text in comment 1, 2 and 3. As for the difference in font size between abstract and text, this was done by the editor. Once more, thanks very much.